# A mixed-methods study of challenges and benefits of clinical academic careers for nurses, midwives and allied health professionals

Diane Trusson ,[1] Emma Rowley,[2] Louise Bramley[3,4]

¹School of Medicine, University of Nottingham, Nottingham, UK
²Business School, University of Nottingham, Nottingham, Nottinghamshire, UK
³Institute of Nursing and Midwifery Care Excellence, Nottingham University Hospitals NHS Trust, Nottingham, UK
⁴University of Nottingham Faculty of Medicine and Health Sciences, Nottingham, UK

**Correspondence to**
Dr Diane Trusson;
diane.trusson@nottingham.ac.uk

## ABSTRACT

**Objectives** The clinical academic trajectory for doctors and dentists is well-established, with research embedded in their career development. Recent years have also seen a burgeoning interest and push for nurses, midwives and allied health professionals (NMAHPs) to pursue a clinical academic career. However, the National Institute for Health Research (NIHR) 10-year review suggested that there may be problems with progression post Master's degree level for this group, with nurses and midwives receiving less NIHR funding than allied health professionals. This study responds to these concerns, tracking the progression and exploring experiences of NMAHPs in the East Midlands region of England.

**Design** An online survey and in-depth interviews were used to capture a wide range of experiences.

**Participants** 67 NMAHPs who were pursuing a clinical academic career were surveyed, supplemented by 16 semi-structured in-depth interviews.

**Results** Three themes emerged during data analysis: Embarking on a clinical academic career, overcoming barriers and benefits.

**Conclusions** NMAHPs are motivated to pursue a clinical academic career by a drive to improve services for the benefit of patients and the National Health Service more widely, as well as for personal development and career progression. People working in these roles have opportunities to explore possible solutions to issues that they encounter in their clinical role through academic study. Findings reveal benefits emanating from the individual level through to (inter)national levels, therefore academic study should be encouraged and supported. However, investment is needed to establish more clinical academic roles to enable NMAHPs to continue to use their experience and expertise post-PhD, otherwise the full extent of their value will not be recognised.

## INTRODUCTION
### Background

There is a long tradition of doctors and dentists pursuing academic research alongside their clinical practice.[1] However, it is increasingly acknowledged that nurses, midwives and allied health professionals (AHPs) (henceforth NMAHPs) are also well-placed to devise solutions to the problems that they observe first

**Strengths and limitations of this study**

► Online survey gathered the views of 67 respondents outside of medicine and dentistry, representing a wide range of occupations and academic achievements.
► In-depth interviews with 16 respondents enabled exploration of issues around career progression and impact resulting from academic study.
► The study was limited geographically to one area of the UK.

hand in their day-to-day clinical practice.[2] A research-active workforce is important to the National Health Service (NHS), which aims to 'build the capacity and capability of our current and future workforce to embrace and actively engage with research and innovation'. (Health Education England, p6)[3] Furthermore, 'the NHS supports and harnesses the best research and innovations to improve patient outcomes, transform services and ensure value for money'.[4p.4] Health Education England (HEE) and the National Institute for Health Research (NIHR) have developed schemes to encourage NMAHPs to pursue postgraduate study in partnerships with the NHS and Higher Education Institutions (HEIs) across the UK, giving them 'the chance to bring their questioning minds, and expertise, to the research table'. (Trueland, p2)[2]

However, in its 10-year report, the NIHR expressed concerns about the 'poor academic progression for non-medical professions from the Masters level', (National Health Service, p2)[5] particularly for nurses and midwives. This contrasts heavily with anecdotal evidence in the East Midlands area of England which suggests good levels of progression achieved by NMAHPs. Previous studies have explored the experiences of doctors and dentists embarking on a clinical academic career,[6 7] yet there is a gap in understanding about the

experiences of NMAHPs. The first regional practitioner network for clinical academics[8] (a joint innovation established between NIHR Collaboration for Leadership in Applied Health Research and Care (CLAHRC) East Midlands and Nottingham University Hospitals NHS Trust), offered an opportunity to close this gap.

The study aimed to track progression of clinical academic NMAHPs in the East Midlands, to explore challenges in combining academic study with clinical practice and to demonstrate the impact on patient outcomes and value of investing in clinical academic careers for NMAHPs.

## METHODS
### Study design
The study had two data gathering components. In the first stage a questionnaire enabled demographic details and progression data to be gathered, with opportunities for free text responses. The second stage of the study used qualitative methodology to enable deeper exploration of experiences from the interviewees' perspective. An interpretive approach was adopted to gather experiences including feelings, emotions and motivation which cannot be measured in an objective way.[9]

This manuscript has been prepared according to the Standards for Reporting Qualitative Research[10] (see online supplementary file 1).

### Research team
The team, who are all experienced researchers, consisted of a medical sociologist, a knowledge translation manager and a clinical academic lead nurse. Authors B and C established the East Midlands Clinical Academic Practitioner Network[8]; author A who conducted all the interviews, had no prior relationship with the target population.

### Context
The research took place in the East Midlands area of England which encompasses 8 acute Trusts, 5 mental health Trusts, 1 ambulance service and 17 clinical commissioning groups.[11 12]

### Recruitment
A link to the online survey[13] was emailed to members of the East Midlands Clinical Academic Practitioner Network[8] and was publicised through social media platforms. Potential respondents were informed that the study aimed to track their progression and identify how they had overcome any challenges along the way. It was emphasised how sharing their experiences could help to ensure smooth progression for future trainees.

The interviewees were self-selected, having indicated their willingness to be interviewed in their survey responses. The rest were recruited through snowball sampling. Although this method has implications for confidentiality and anonymity, it was an effective way of identifying individuals whose experiences were relevant to the research.[14]

### Ethical considerations
Under the guidance provided by the Health Research Authority,[15] ethical approval was not required for this study because participants were recruited by virtue of their participation in educational programmes, rather than their NHS status. Nevertheless, good research governance was observed, that is, information was provided to participants and verbal consent was obtained prior to each interview. Participants were made aware of their right to withdraw from the study, assured that any data published would be anonymised and that data would be stored confidentially on secure university systems.

### Data collection methods
A Bristol Online Survey[13] was created (see online supplementary file 2), aiming to:
I.   Track the progression of clinical academics across the pathway, and
II.  Explore the ways in which training programmes and clinical academic roles had impacted on the respondents' clinical practice.

The survey was open for a 2-month period. This was followed by 16 semi-structured, in-depth interviews.

### Data collection instruments
The survey and interview topic guide (see online supplementary file 3) were designed by the research team. Interviews began with an open question inviting participants to describe their experiences of being a clinical academic followed by a series of questions aimed at exploring changes to clinical practice as a result of academic training, the impact of their research and any influence on their colleagues. The topic guide enabled cross-case comparability while the semi-structured format enabled flexibility with the order of the questions so that topics could emerge naturally through the interview and allowing participants to elaborate and give examples to support their answers.[14] Participants were also invited to add further comments at the end of the interview. Each interview lasted around 1 hour; they were digitally recorded with participants' consent. Details of the interview sample are shown in table 1.

### Interview sample

### Data processing
Following professional transcription, data were anonymised with all identifying aspects removed from the transcripts prior to analysis. In the results section below, participants are identified by their interview case study (CS) or survey respondent (SR) number and professional group only, to preserve anonymity.

### Data analysis
Survey responses were collated and summarised using the Bristol Online Survey[13] software. Descriptive numerical data were represented in graphs and tables.

Qualitative data from the free-text survey responses and the interview transcripts were combined and analysed using

| Table 1 | Interview sample characteristics | | | | |
|---|---|---|---|---|---|
| Case study | Clinical role | Stage of study (at interview) | PhD year | Age group | Gender |
| 1 | Nurse/midwife | Thesis pending | | 20–30 | Female |
| 2 | Nurse/midwife | PhD | 2 | 41–50 | Female |
| 3 | Nurse/midwife | PhD | 4 | 41–50 | Female |
| 4 | Nurse/midwife | Postdoc | | 41–50 | Female |
| 5 | Nurse/midwife | PhD | 3 | 41–50 | Female |
| 6 | AHP | PhD | 4 | 41–50 | Female |
| 7 | AHP | PhD | 2 | 41–50 | Female |
| 8 | Nurse/midwife | PhD | 5 | 41–50 | Female |
| 9 | Nurse/midwife | Postdoc | | 31–40 | Male |
| 10 | AHP | PhD | 2 | 41–50 | Female |
| 11 | AHP | PhD | 1 | 41–50 | Female |
| 12 | AHP | PhD | 4 | 51+ | Male |
| 13 | AHP | PhD | 3 | 31–40 | Male |
| 14 | AHP | PhD | 3 | 41–50 | Male |
| 15 | AHP | Postdoc | | 51+ | Male |
| 16 | AHP | PhD | 1 | 31–40 | Male |

AHP, allied health professional; Postdoc, postdoctoral.

thematic analysis. This involved reading through the data to manually identify 'patterned responses or meaning within the data set'. (Braun, p82)[16] Themes which arose iteratively from the data, were discussed and agreed by the research team which enhanced trustworthiness of the analysis.[14]

### Rigour

All three authors were involved in the research design and in data verification throughout the data collection and analysis processes to assure quality and rigour.

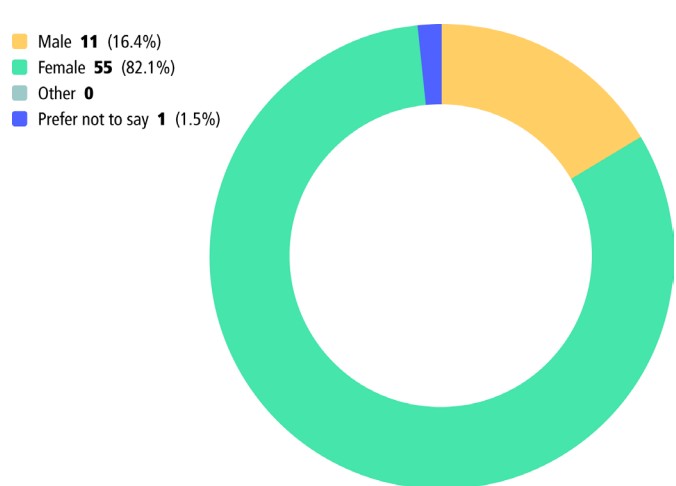

Male 11 (16.4%)
Female 55 (82.1%)
Other 0
Prefer not to say 1 (1.5%)

**Figure 1** How participants described their gender.

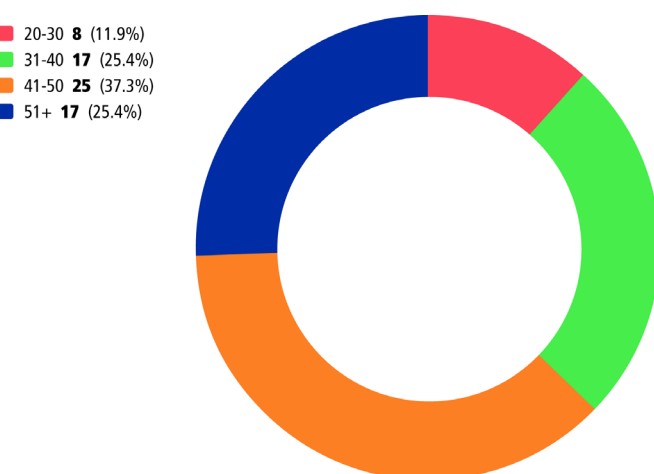

20-30 8 (11.9%)
31-40 17 (25.4%)
41-50 25 (37.3%)
51+ 17 (25.4%)

**Figure 2** Survey respondents' age groups.

## RESULTS

### Respondents

There were 81 responses to the online survey; however 14 responses were excluded because they were not members of the East Midlands Clinical Academic Practitioner Network. This is a limitation of using social media to publicise the survey.

The following sections report the findings from a combination of survey and interview data with a particular focus on the academic pathway. Three major themes are discussed: embarking on a clinical academic career, overcoming barriers and benefits of clinical academic research. Within this discussion are issues around funding, management support, impact and encouraging future clinical academic leaders.

### Gender

The gender split of participants (see figure 1) was expected, due to the predominantly 'female gendered' occupations being questioned. For example, the majority of nurses (who outnumber AHPs considerably) and midwives, identify as female.[17] There were 26 respondents in the study who held these roles.

### Age

Despite the HEE aim of producing future clinical academic leaders early in their career,[3] results in figure 2 illustrate that funding is currently being used to support individuals at mid/late career stage. Consequently, there may be potential implications for the career level (progression) and impact NMAHP clinical academics are able to achieve before they reach retirement age. This contrasts to the rhetoric of investing in future leaders, while also suggesting that 'the potential of high-achieving graduates is underexploited' (Baltruks, p8)[18] as participants were waiting some years post-undergraduate award, to pursue clinical academic ambitions.

The current culture within the clinical setting was described as a particular barrier for clinical academic progression by participants in the 20 to 30 year age bracket:

I have come across lots of negativity in pursuing a clinical academic career as a nurse who is only a few years qualified (SR13 nurse/midwife).

They think to be an expert in your field you must've been qualified for like fifteen plus years. Well that's just ridiculous because I know a lot more than some colleagues who've been working double the amount of time that I have and that's just because I like to understand why I'm doing what I'm doing (CS1 nurse/midwife).

This indicates a need for a culture change so that NMAHPs are supported to join the clinical academic career trajectory at an earlier stage, for instance through the apprenticeship model.[19] It seems that in some cases, individuals are expected to have a number of years' experience and a secure clinical role before they are supported to embark on the clinical academic pathway, rather than being able to develop parallel roles like their medical colleagues.

### Embarking on a clinical academic career
#### Motivation
In contrast to doctors and dentists, research has not traditionally been a career route for NMAHPs. Participants described being self-motivated to pursue research, rather than follow a predefined path:

I've always, from very early on in my clinical career, had an interest in evidencing the work that I was doing. So I self-motivated really, did service audits and evaluations (CS14 AHP).

Participants overwhelmingly described how their interest in research was driven by improvements to patient care:

Clinical academics are part of the solution. We can innovate and generate the solutions for these age-old problems that we're seeing, having a robust methodological approach to understanding and exploring the phenomena. But also developing and testing interventions to address these problems (CS9 nurse/midwife).

This illustrates the potential value of investing in clinical academic careers for NMAHPs who can move change from an idea into a tested intervention within their clinical practice.[20]

Participants often considered leadership of other people in their decision to pursue academic study:

I want to do this for me, but I also want to do it for my daughters to show that women can be in science and can lead in these fields and yes we might have to juggle family things and children, but you can do it (CS7 AHP).

Undertaking doctoral training and following a clinical academic career was also perceived to be about leadership opportunities and potential, as well as building a culture within the NMAHP professions so that many others could follow the path started by the few.

It's essential for me to have a PhD because we need people to mentor, to supervise. I need to be at that level for the staff coming through, to help them (CS8 nurse/midwife).

### Overcoming barriers
#### Funding
The survey respondents included 35 current PhD students and 10 postdoctoral NMAHP clinical academics. Of these, 11 respondents' PhDs were funded by the NIHR, 2 of which were the highly competitive and prestigious NIHR Clinical Research Doctoral Fellowship:

With a mortgage, a baby, one on the way it was only an NIHR fellowship… it was that or nothing (CS13 AHP).

I'm the main breadwinner, I earn more than my husband … so that financial part was a big barrier for me. I knew that the best financial support were the NIHR ones, so I took that time to develop that application. It didn't just affect me, it would affect the whole family (CS7 AHP).

Students who did not manage to get NIHR funding were likely to be offered standard UK Research and Innovation stipends (approximately £14.5K per year).[21] This was problematic for participants who had often reached high pay bands by the time they embarked on an academic pathway (eg, the £40K+per year salaries for a senior band 7/8 practitioner).[22] It also meant they faced tricky decisions and negotiations about pension and employment rights:

Clearly I think an obstacle is when you get to that high clinical level and you've got mortgages and things, it makes it very difficult to do it on a basic stipend (CS14 AHP).

Some participants had received financial support from their employer which enabled them to make the move into academic study:

My employers said that if I applied for a stipend and was successful they would top me up to my full salary which would allow me to do it essentially full-time (CS14 AHP).

However, this experience was not the norm; rather, NMAHPs were being forced to make sacrifices in order to develop their career portfolio and skills. Nevertheless, 23 respondents (including 11 who had progressed from the NIHR funded Master's degree in Research Methods) secured PhD funding from alternative sources as indicated in figure 3 below:

These 23 respondents included 11 nurses and 1 midwife which is interesting in terms of the NIHR 10-year report which says that **nurses and non-healthcare professionals are less successful** and, 'For ICA, the **lower success for**

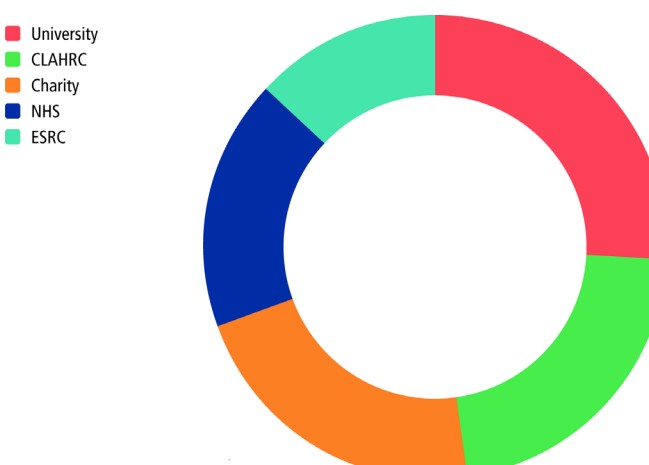

University
CLAHRC
Charity
NHS
ESRC

**Figure 3** Source of PhD funding (non-NIHR): CLAHRC = Collaboration for Leadership in Applied Health Research and Care; NHS = National Health Service, ESRC = Economic and Social Research Council

**nurses is a particular concern**'(National Health Service, p11)[5] (emphasis added). Our (regional) findings suggest that nurses are successful in securing alternative funding for their PhD studies, therefore NIHR metrics may not be appropriately capturing all NMAHP progression. By restricting progression metrics to purely NIHR funding sources, they are providing a narrative that does not reflect the experiences of clinical academics from these occupations working across the East Midlands region.

### Maintaining a clinical role

The NIHR's guide to the Integrated Clinical Academic (ICA) Programme describes how 'the individual's academic and clinical 'jobs' are not mutually exclusive but are instead complementary, informing and supporting each other, and definable within a single role'. (NHS Health Education, p6)[23] Participants described the benefits of working in their clinical setting:

> When I was on the ward I could kind of forget about the PhD but also recognise how it was shaping me as a nurse (CS1 nurse/midwife).

However, working at the same time as doing a PhD was identified as challenging. One participant who had secured a prestigious post before completing her PhD said:

> The promotion is massive and the PhD is hugely important. You've got to somehow survive with the work and academia all at once and not fall down the rabbit hole and get lost. Yeah that's a big challenge (CS8 nurse/midwife).

Recognising this challenge, the Council of Deans of Health recommend 'support from the clinical side including agreed study time (Baltruks, p9)[18] for clinical academics. However, a recurring theme in the data was that managers are often dealing with operational care delivery challenges and were unwilling, or unable, to release staff to do research:

> You have to get your line manager to sign the application form to say they'll support you and it took a lot of effort to get that signed. They only gave in because my contract was 22½ hours a week. They said 'what you do in the rest of the time is your own business, but it can't impact on this, we're not giving you any time off'. It was 'what's this got to do with your job?' (CS5 nurse/midwife).

A common, potentially problematic issue encountered by the respondents was an apparent lack of recognition of the value of research:

> There's a huge untapped workforce…with the right support and time we could be doing things more effectively and more efficiently, but that isn't necessarily valued in organisations. We've got to see this many patients, (we're) not using our skills of criticality, reflectivity; we're not going to innovate and change practice (CS7 AHP).

Despite their achievements during the PhD, many participants expressed anxieties about their future careers, having been made to move aside clinically in order to progress their academic ambitions, rather than being able to develop their academic and clinical skills in tandem. For example a dietician said:

> Recently I've had to step out of my area of expertise… I'm just doing general, allergies, weight management, which is not my area, but I need to pay the mortgage (CS10 AHP).

This indicates a serious lack of organisational and professional value placed on the knowledge and skills achieved by some clinical academics. As a result, some participants felt there might be no option to stay in their clinical role post-PhD:

> I would be keen to stay more NHS-based but constraints with funding and time might end up pushing quite a few of us out into university (CS2 nurse/midwife).

> I currently work for an NHS trust, but the lack of support makes me wonder if the only option is to not work clinically, or work bank/agency, which to me is not embracing the value clinical academics can bring to the clinical area (CS13 nurse/midwife).

In contrast to the clear career trajectory for doctors and dentists, 'early career clinical academics face uncertain career paths and may choose the comparably stable worlds of clinical practice where their skills are in high demand, or a dedicated academic career.' (Health Education England, p9)[3] The main reason being; 'the scarcity and highly competitive nature of NMAHP postdoctoral research positions (…) and the comparative lack of research funding for healthcare professions other than

medicine' (Health Education England, p9)[3] as this participant articulated:

> My frustration (is that) the pathway is a pyramid therefore some people will not progress up (CS12 AHP).

The participants who had successfully negotiated this 'pyramid' generally did so with the help and support of key individuals within the organisation who were able to champion the cause of aspiring clinical academics at board level. For example:

> I have a very supportive divisional head nurse and have been appointed into a trailblazer post; we haven't got anything similar within the organisation. So there's real potential to forge out innovative ways in which clinical academics can fulfil that remit of working in clinical practice and undertaking research, but also pave the way for others that want to come up (CS9 nurse/midwife).

Similarly, the chief medical officer argues that 'developing the next generation of research leaders in clinical research is essential to the UK, (Davies, 502)[24] but the data suggest a cliff-edge in the pathway; greater numbers of practitioners are embarking on clinical academic careers, but following PhD, opportunities are scarce. The fortunate have the organisational backing to 'trailblaze', but others face a decision to return to their pre-PhD clinical role (and hence not have their academic skills recognised and utilised) or follow a traditional academic research pathway and leave their clinical post behind (thus negating the whole reason for pursuing a clinical academic career).

### Benefits and impact

The data revealed multiple levels of impact and the value of academic research as the ecological model[25] in figure 4 illustrates:

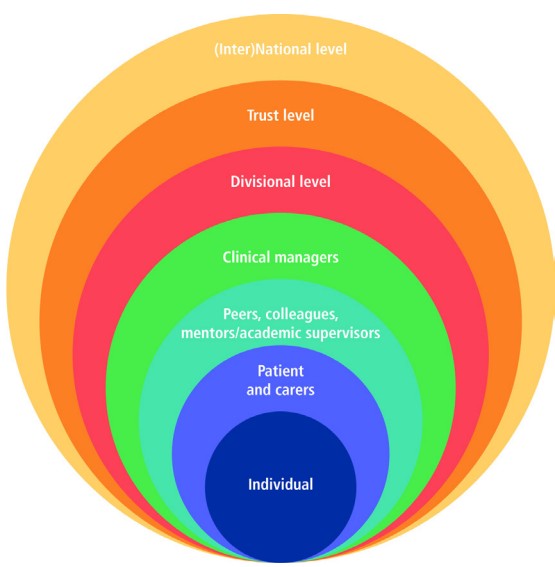

**Figure 4** Ecological model.

### Individual level

Participants reported benefits such as job satisfaction, increased awareness of research, enhanced skills and sense of achievement. Progression post-master's level was achieved by 52 respondents, of whom 19 were nurses/midwives. Although this echoes concerns for the relatively low progression rates of nurses/midwives when compared with AHPs,[5] 6 nurses/midwives were pursuing Master's level courses at the time of the survey and may have since progressed.

The clinical academic pathway had presented opportunities for career progression. One participant had been 'talent-spotted' and offered 'numerous extra employment and development opportunities'. When asked about any changes to their employment/role/grade since embarking on the clinical academic pathway, almost half (31) of the 63 survey responses described positive changes, although 5 of these reported no change in their pay band. Six had been seconded but risked reverting to their lower pay band when the secondment ended.

Respondents who had gained clinical academic roles described having separate contracts of employment (ie, one each for NHS and academia) with separate employment and pension rights. Nine respondents had been promoted with one becoming a consultant midwife, and another achieving a clinical lectureship (a pivotal post in the NIHR ICA funded clinical academic trajectory).

### Patients and their carers

The research undertaken by the participants could potentially make a big difference to patient outcomes and experiences. For example, an AHP had identified an element of practice that could extend and improve the lives of seriously ill children. Another participant had introduced a pre-surgery exercise programme which helped patients to feel involved in the process and was highly rated in a patient satisfaction survey.

These examples illustrate how NMAHPs on a clinical academic pathway are uniquely placed to develop interventions which can be easily implemented with positive results for patient care.

### Peers/colleagues

Many participants described their pride at becoming role models, able to support other colleagues into academic study:

> I've mentored a lot of different kinds of professions to actually realise that it is doable. That's a really rewarding side to the job to think that you might have helped somebody (to) be a more able clinician, a more able academic which can then impact on the patient (CS7 AHP).

This demonstrates the important role of mentors in encouraging future clinical leaders by providing 'pastoral support and help(ing) mentees deal with the demands of a clinical academic research career'. (Baltruks, p10)[18] It

also reveals the value of investing in staff who then give back.

## NHS Trust

Recognising that research intensive organisations have better patient outcomes, the Care Quality Commission have introduced research into its quality framework.[26] Respondents described numerous benefits for the NHS organisations who supported clinical academic careers. For example:

> At the hospital they want this Magnet status.[27] The three domains are good clinical outcomes, patient experience and staff experience and part of (that) is having well qualified nurses. They really want to increase the academic underpinnings of nurses and have research leaders…what I'm doing really ticks the boxes of Magnet (CS5 nurse/midwife).

Participants highlighted how supporting clinical academic careers could address current issues with recruitment and retention:

> Forty thousand nurses we have a national deficit of, so people can choose where they want to work. They'll be looking for organisations that are aspirational. So actually offering innovative career pathways that can intellectually challenge, but also have that direct patient care element, is going to be attractive to a lot of people (CS9 nurse/midwife).

These comments resonate with the NIHR's advice for aspiring clinical academics 'to base themselves within organisations where the importance of research is well understood and clinical academic careers are appropriately supported.' (NHS Health Education, p6)[23]

The data revealed numerous examples of impact resulting from participants' clinical academic careers, including the potential for substantial savings. For example, one participant's intervention removes the need for GPs' referral for physiotherapy, potentially saving 'multimillion pounds' across the NHS, and has subsequently been recognised in the NHS long-term plan.[28] This illustrates how clinical academics can develop 'well informed and relevant research' (NHS England, p5)[4] that can quickly be transferred into practice for the benefit of the NHS.

## Inter(national)

Participants provided details of multiple academic journal articles and conference presentations enabling worldwide dissemination of their research. In addition, one participant was invited to join an International Working Party developing consensus guidelines for treating children with kidney disease.

In another example, an open-access resource to help professionals to deal with children in mental health crises, specifically those at risk from self-harm, had been 'disseminated nationally, not just within health, but also in social care and education settings'. This illustrates the wide-ranging impact of clinical academics' research which can occur more speedily than traditional research.

These reported experiences represent a small snapshot of the benefits of supporting NMAHPs to pursue clinical academic careers and the need to do so, as one participant articulated:

> Moving forward we have to look at more sustainable and integrated approaches to embedding clinical academic careers. I'm excited to hear that there's an apprenticeship framework coming out because for clinical academic careers to be truly embedded within non-medical professional career pathways, it has to be driven by the NHS. Universities get the value of clinical academics and they're on board, but for it to truly work, we need to have change within the NHS (CS9 nurse/midwife).

## DISCUSSION

This study has explored the experiences of NMAHPs in the context of the NIHR 10-year report which expressed concern about the 'poor academic progression for non-medical professions from the Masters level'. (National Health Service, p2)[5] Our data reveal that NMAHPs do progress post-Masters; although through alternative means than the NIHR pathway. In addition, the findings indicate good levels of career progression post-PhD, including progression into consultant midwife and clinical lectureship fellowship roles, and the far-reaching impact of research. Success was achieved despite barriers such as a lack of organisational and managerial support for NMAHPs wanting to pursue a clinical academic career path. Although operational challenges to care delivery are to be acknowledged, this suggests a need for a change of culture in line with NHS recommendations for a research-active workforce, especially in under-represented roles such as nurses and midwives.[5]

The results of this study confirm that AHPs are more likely than nurses/midwives to progress post-Master's degree.[5] One possible explanation is that there may be some association with the length of time that AHPs have been an all degree profession whereas until relatively recently, only a small proportion of nurses graduated with a degree.[29] Once the degree pathway becomes more embedded and clinical academic role models more common, nurses may make earlier career decisions to engage with research training, and choose to follow a clinical academic trajectory, rather than management or specialist services which are the more traditional career pathways. Also of concern is the relatively late age of NMAHPs embarking on the clinical academic pathway. To address this, the Council of Deans recommend that all health professionals are exposed to the benefits of health research at undergraduate level[18] and encouraged to embark on a clinical academic pathway earlier in their careers. This aims to promote a culture where research is the norm, rather than the exception, supporting the

HEE aim of 'raising the profile of research and innovation among the potential future workforce as an integral part of all healthcare roles'. (Health Education England, p10)[3] A recurrent theme among NMAHPs who had achieved success was having champions and role models who mentored them and were able to promote the benefit of supporting clinical academics at the Executive Board level. The national Clinical Academic Roles Development Group recommend mentorship as an enabler of success[20] which this study's findings support, while also showing how current clinical academics mentor their junior colleagues. Critical mass is important and therefore it is essential that aspiring NMAHP clinical academics bring others with them.

Like previous research, this study's findings suggest a disconnect between priorities at senior management level and 'what happens on the ground'. (Springett, p39)[30] To combat this, the Clinical Academic Roles Development Group recommends that 'evidence of the link between clinical academic roles and improved outcomes/patient benefit/research activity etc' is vital for securing ongoing investment in clinical academic careers. (Association of UK University Hospitals (AUKUH), p18)[20] This study has demonstrated how NMAHPs' innovative research has potential to increase efficiency, with potential cost-benefits for the NHS, as well as benefitting HEIs through grants and publications they generate, providing a bridge between tensions in priorities of the NHS (efficient services and improved patient outcomes) and HEIs (grants income generation and publications).[29] It follows that close partnerships between NHS and HEI organisations should be developed following Southampton's model, where NHS and HEI partners have successfully collaborated to build academic pathways and increase research capacity.[1 20] Encouraging a research-active workforce is also important for recruitment and retention of 'highly motivated clinicians who often become the leaders of tomorrow' (Association of UK University Hospitals (AUKUH), p11).[20]

NMAHPs are committed to conducting high-quality research alongside their clinical role. However, pursuing a clinical academic pathway was likened to a pyramid where progression becomes increasingly challenging. In contrast to their medical colleagues whose medical and academic training occurs in tandem, many NMAHP participants undertaking PhDs were faced with the prospect of having no job to return to, or taking pay cuts or reduced hours. The findings indicate an urgent need for a clinical academic pay scale, whereby clinical and academic skills are valued equally by NHS and HEI organisations, and a coherent career pathway to be developed for NMAHPs post-PhD, where research experience is valued and utilised.[20] Having a job description which enables research within a clinical role would avoid the dilemma faced by some participants in this study who were considering a purely academic career with a resultant loss of expertise for the NHS. Similarly, the Council of Deans recommend allocating funding and mentoring

to enable clinicians with doctorates to develop both sets of skills; pointing out that 'a PhD is the 'end of the beginning' of research training rather than an end in itself'. (Baltruks, p10)[18]

## Strengths and limitations

The mixed methods used in this study enabled data collection from a wide range of health professionals using strengths of different approaches[14] to create a fuller picture of NMAHPs' experiences of embarking on a clinical academic pathway. Although limited in number and by geographical location, the data provide useful insights into the experiences of this under-researched group and provide a foundation for future studies of NMAHPs' experiences in other locations. A further limitation is that although comparisons were made with medical colleagues, their experiences were not sought for this study. However, a comparative study with medical clinical academics in the East Midlands is currently underway.

## CONCLUSION

This study has discussed NMAHPs' motivations for embarking on a clinical academic pathway and how challenges were overcome along the way. It shows how investing in clinical academic training for NMAHPs is vital in developing and retaining a research-active workforce where patient care remains the central consideration. However, in order to do this, there needs to be support at all organisational levels to enable NMAHPs to engage in developing a clinical academic career, and preferably at an earlier stage in their career development. In addition, investment is needed to establish more clinical academic roles post PhD. The Clinical Academic Roles Development Group (formally hosted by and known as the AUKUH group) have produced a range of recommendations in this respect, along with concrete examples of successful implementation of clinical academic posts from across the UK[20] Developing and facilitating a clear clinical academic career path will ensure that both the clinical and research expertise and experience of clinical academic NMAHPs continues to be utilised fully for the benefit of patients and the NHS as a whole.

**Contributors** All authors (DT, ER and LB) were involved with the study design and data analysis. DT collected the data and drafted the manuscript as lead author. ER and LB contributed comments and edits to the manuscript. Final approval was given by all authors.

**Funding** This work was supported by the National Institute for Health Research (NIHR) Collaboration for Leadership in Applied Health Research and Care East Midlands (CLAHRC EM), grant number S-CLA-0113-10014. The views expressed are those of the authors and not necessarily those of the NIHR or the Department of Health and Social Care.

**Competing interests** None declared.

**Patient consent for publication** Not required.

**Provenance and peer review** Not commissioned; externally peer reviewed.

**Data availability statement** Data are available upon reasonable request.

**ORCID iD**
Diane Trusson http://orcid.org/0000-0002-6995-1192

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
