## [Reviewer comments · BMJ Open]

ARTICLE DETAILS

TITLE (PROVISIONAL)	A mixed-methods study of challenges and benefits of clinical academic careers for nurses, midwives and allied health professionals
AUTHORS	Trusson, Diane; Rowley, Emma; Bramley, Louise

VERSION 1 – REVIEW

REVIEWER	Professor Sheree Smith Western Sydney University, Sydney, Australia have written on clinical academic careers from the Australian perspective and a blog of the Council of Deans of Health UK
REVIEW RETURNED	18-Apr-2019

GENERAL COMMENTS	Thank you for this manuscript which I found interesting given the investment by NIHR in awards for registered nurses and midwives to advance their clinical research careers for nearly two decades. This study missed an opportunity to reveal the differences in foundational professional learning as medicine (MBBS) and allied health (physiotherapy) has ensured research training commences at the entry to the professional degree level both within the UK, Europe, North America and the Asia Pacific region. Given UK RNs professional programs have only recently moved to a bachelor level (although the curriculums may or may not have changed) and is one of the last developed countries to do so, this may impact of opportunities for early career decisions by RNs to engage with research training rather than management or specialist service which are the traditional career pathways. Another key area missing is an understanding of potential mentorship by those senior academics in named Chair positions and their contribution to support of the NIHR fellow as an academic and organisational champion. This manuscript starts an important conversation which could have been explored further however as it is written it is to be valued. From a research design perspective, the notion of trustworthiness of qualitative data is not reported, type of analysis is not described, limitations appear not to be considered and the COREQ or SRQR checklist was not apparent as a supplemental file. The other concern is the research ethics statement in the manuscript is lacking in details. Whilst purposive sampling is common as is the snowball technique, better reporting of what information potential participants had, how they were approached, confidentiality around these processes and were they informed in the first phase that they may be contacted again and the confidentiality issues around snow ball technique need to be conveyed to the reader.
--

	I would like to re-state, this manuscript is well written and starts important conversations around preparatory professional education, organisational understanding and support, and human investment in a clinical research career for registered nurses and midwives.
--	--

REVIEWER	Linda Tinkler Newcastle upon Tyne Hospitals NHS Foundation Trust And University of Sheffield, School of Nursing & Midwifery, UK
REVIEW RETURNED	03-Jun-2019

GENERAL COMMENTS	Thank you for submitting this paper, I enjoyed reading it very much. I recognise and can relate to a lot of the issues raised and discussed and was very interested by your findings. This is an extremely important area and I congratulate you for sharing this work to add to the evidence base to benefit us all. I do have some comments that I hope will be helpful to you: In describing a piece of qualitative research I would expect to see reference to the COREQ framework for reporting. I know this may not always be appropriate depending on the methodology and the stance you take and I am not sure if this journal requires it, but it is a helpful tool for ensuring the main aspects of the research are reported. I felt that some of the items on the COREQ were missed, so it may be worth reviewing that for structuring your work. In the background section you cited concerns voiced in the NIHR ten year review with regards to Nurse and Midwives attracting less funding than AHPs post Masters level, which you were planning to respond to by tracking progression in your locality alongside exploring NMAHP research experiences. I feel there was a missed opportunity to share or at least describe broadly the progression data - you mentioned descriptive statistics, graphs and charts but there are none to see? Not sure if I missed a file somewhere but I can't see anything other than the demographics, and ecological model. I am left wondering what progression looked like for you as it seems the findings focuses on qualitative themes from the interviews and free text responses to the survey. You describe some of the progression data in the discussion section but this then comes as a surprise.... Are you able to share the questions that you asked as part of the online survey? Is there a topic guide for the interviews? Is there a coding tree of how you developed your themes? Did you do this manually or with a piece of software? This would help the reader decide if the questions you asked were relevant to the findings you are describing, and will also enable them to reproduce the study in their area if appropriate. I am not familiar with the term "Ethical clearance" not sure if it is a geographical term but perhaps the terminology could be clearer in terms of ethical review and why it was not required. When you describe "Good Research Governance" what did that look like? Was informed consent written? Thank you for sharing the demographic data, I was interested with the points you highlighted about age groups and the importance of supporting NMAHPs from earlier in their careers, this is a
--

	challenge that we need to do more work on and could have been drawn out much more in the discussion. I found the results very interesting and I like how they have been described and set out, however I am not sure I am comfortable with the title of one theme "coming to the clinical academic career?" Is this not about people's drive, passion, determination and persistence in developing/pursuing a clinical academic career? "Coming to" feels a little passive in comparison to the experiences and journeys of clinical academics and it doesn't fit for me (sorry!) I also wonder (and this is very minor) if the drive to improve patient care should come first in that section, as this is usually what sparks an initial interest in pursuing a specific research question, rather than career progression first? Not sure if that's what you got from your data so I'll let you decide on that one though! I enjoyed reading the section on barriers, funding is a big aspect and I like how you have articulated those challenges, again we still have some way to go with this to smooth the path for our NMAHPs. I wasn't a fan of the choice language in one of the quotes in the section on maintaining a clinical role if I'm honest, but if that's what your participant said....! I am assuming this is someone who has finished the PhD study/funding and is still writing up but has returned to clinical practice? Again I liked the section on career post PhD as sustainability is a challenge for us and I'm very pleased to see this articulated in the literature here... I very much enjoyed reading the benefits and impact section, however I felt the discussion section lacked debate and missed the opportunity to draw on other relevant literature. I felt the discussion was a little descriptive to be honest and this was where some of the progression data was introduced which came as a surprise after the findings focussed on the qualitative data? I wonder if you could take a look at this section and reference a little more literature and bring in some debate around NHS culture generally and perhaps supporting those earlier in their careers as identified earlier in the paper to draw this all together? I also felt the conclusion didn't accurately summarise/reflect what you described in your paper and missed the opportunity to make some bold suggestions for what we need to do next! I hope the comments above are helpful to you, I really enjoyed reading your paper, we need more literature like this out there to support our work in this pathway....I look forward to seeing it published.
--	--

REVIEWER	Greta Westwood Florence Nightingale Foundation UK
REVIEW RETURNED	08-Jun-2019
GENERAL COMMENTS	Good work adding to the body of literature providing the evidence that the clinical academic pathway for NMAHPs needs urgent attention to be equivalent to that of the medical CA pathway

VERSION 1 – AUTHOR RESPONSE

Reviewer 1.

R1 Comment:

Thank you for this manuscript which I found interesting given the investment by NIHR in awards for registered nurses and midwives to advance their clinical research careers for nearly two decades. This study missed an opportunity to reveal the differences in foundational professional learning as medicine (MBBS) and allied health (physiotherapy) has ensured research training commences at the entry to the professional degree level both within the UK, Europe, North America and the Asia Pacific region.

Given UK RNs professional programs have only recently moved to a bachelor level (although the curriculums may or may not have changed) and is one of the last developed countries to do so, this may impact of opportunities for early career decisions by RNs to engage with research training rather than management or specialist service which are the traditional career pathways.

Response:

Thank you for prompting us to discuss this important difference in the provision of research training for different professionals. We feel that the discussion section has been enhanced by considering this issue as a reason why nurses may be lagging behind their AHP colleagues when it comes to pursuing a clinical academic career.

We will be comparing NMAHPs' experiences with those of medical clinical academics in a later paper when results from an ongoing comparative study have been analysed. This has been highlighted in the limitations section.

R1 Comment:

Another key area missing is an understanding of potential mentorship by those senior academics in named Chair positions and their contribution to support of the NIHR fellow as an academic and organisational champion.

Response:

We are grateful for this suggestion. The role of mentors was an important factor in NMAHPs' success so this has now been added to the results and discussion sections.

R1 Comment:

This manuscript starts an important conversation which could have been explored further however as it is written it is to be valued.

Response:

Thank you for this encouragement. The issues raised in this study will be further explored when comparing experiences of NMAHPs and medical clinical academics.

R1 Comment:

From a research design perspective, the notion of trustworthiness of qualitative data is not reported, type of analysis is not described, limitations appear not to be considered and the COREQ or SRQR checklist was not apparent as a supplemental file.

Response:

The methodology section has been revised in line with the SRQR checklist which has been attached.

R1 Comment:

The other concern is the research ethics statement in the manuscript is lacking in details.

Response:

We have provided more details of the steps taken to address ethical considerations.

R1 Comment:

Whilst purposeful sampling is common as is the snowball technique, better reporting of what information potential participants had, how they were approached, confidentiality around these processes and were they informed in the first phase that they may be contacted again and the confidentiality issues around snowball technique need to be conveyed to the reader.

Response:

This section has been expanded to explain the sampling techniques used in the study. We describe the information that was provided to participants and how they had the opportunity to volunteer for follow-up interviews and also discuss the pros and cons of snowball sampling.

R1 Comment:

I would like to re-state, this manuscript is well written and starts important conversations around preparatory professional education, organisational understanding and support, and human investment in a clinical research career for registered nurses and midwives.

Response:

Thank you for these encouraging comments.

Reviewer 2

Thank you for submitting this paper, I enjoyed reading it very much. I recognise and can relate to a lot of the issues raised and discussed and was very interested by your findings. This is an extremely important area and I congratulate you for sharing this work to add to the evidence base to benefit us all.

I do have some comments that I hope will be helpful to you.

Response:

We value the reviewer's enthusiasm and feedback on the manuscript.

R2 Comment:

In describing a piece of qualitative research I would expect to see reference to the COREQ framework for reporting. I know this may not always be appropriate depending on the methodology and the stance you take and I am not sure if this journal requires it, but it is a helpful tool for ensuring the main aspects of the research are reported. I felt that some of the items on the COREQ were missed, so it may be worth reviewing that for structuring your work.

Response:

We are grateful for this suggestion and agree that it helps to structure the manuscript. The SRQR framework (as an alternative to COREQ) has been followed and is attached as requested by the journal editor.

R2 Comment:

In the background section you cited concerns voiced in the NIHR ten year review with regards to Nurse and Midwives attracting less funding than AHPs post Masters level, which you were planning to respond to by tracking progression in your locality alongside exploring NMAHP research experiences. I feel there was a missed opportunity to share or at least describe broadly the progression data - you mentioned descriptive statistics, graphs and charts but there are none to see? Not sure if I missed a file somewhere but I can't see anything other than the demographics, and ecological model. I am left wondering what progression looked like for you as it seems the findings focuses on qualitative themes from the interviews and free text responses to the survey. You describe some of the progression data in the discussion section but this then comes as a surprise....

Response:

Thank you for this suggestion. We have added a chart (figure 3) which shows how 23 respondents had achieved progression post PhD by securing funding from alternative sources.

We have also included further progression data in the results section where the benefits at an individual level are presented.

R2 Comment:

Are you able to share the questions that you asked as part of the online survey?

Is there a topic guide for the interviews? Is there a coding tree of how you developed your themes?

Did you do this manually or with a piece of software? This would help the reader decide if the questions you asked were relevant to the findings you are describing, and will also enable them to reproduce the study in their area if appropriate.

Response:

The survey and interview topic guide are attached, and we have also provided a more detailed discussion of the interview questions.

In the section on analysis, we describe how themes were identified manually and agreed between the research team to ensure rigour.

R2 Comment:

I am not familiar with the term "Ethical clearance" not sure if it is a geographical term but perhaps the terminology could be clearer in terms of ethical review and why it was not required. When you describe "Good Research Governance" what did that look like? Was informed consent written?

Response:

The term ethical clearance has been removed and this section has been re-worded for clarity. In addition, we have provided a fuller explanation of the ways in which ethical considerations were addressed.

R2 Comment:

Thank you for sharing the demographic data, I was interested with the points you highlighted about age groups and the importance of supporting NMAHPs from earlier in their careers, this is a challenge that we need to do more work on and could have been drawn out much more in the discussion.

Response:

Thank you for this suggestion. A much more detailed discussion around the issue of age and career stage of NMAHPs embarking on the CA pathway has been included in the discussion section.

R2 Comment:

I found the results very interesting and I like how they have been described and set out, however I am not sure I am comfortable with the title of one theme "coming to the clinical academic career?" Is this not about people's drive, passion, determination and persistence in developing/pursuing a clinical academic career? "Coming to" feels a little passive in comparison to the experiences and journeys of clinical academics and it doesn't fit for me (sorry!)

Response:

We agree and have amended this title to 'embarking on a clinical academic career' to better reflect the driving forces in developing and pursuing a clinical academic career as you have identified.

R2 Comment:

I also wonder (and this is very minor) if the drive to improve patient care should come first in that section, as this is usually what sparks an initial interest in pursuing a specific research question, rather than career progression first? Not sure if that's what you got from your data so I'll let you decide on that one though!

Response:

This section has been reworked to show that a drive to improve patient care was indeed the participants' main motivation as articulated in the data.

R2 Comment:

I enjoyed reading the section on barriers, funding is a big aspect and I like how you have articulated those challenges, again we still have some way to go with this to smooth the path for our NMAHPs.

Response

We are pleased that this message has come across.

R2 Comment:

I wasn't a fan of the choice language in one of the quotes in the section on maintaining a clinical role if I'm honest, but if that's what your participant said....! I am assuming this is someone who has finished the PhD study/funding and is still writing up but has returned to clinical practice?

Response:

This section has been removed and a fuller explanation given of this participant's circumstances. This has been done without losing the essence of the quote, but avoids causing offence.

R2 Comment:

Again I liked the section on career post PhD as sustainability is a challenge for us and I'm very pleased to see this articulated in the literature here...

I very much enjoyed reading the benefits and impact section, however I felt the discussion section lacked debate and missed the opportunity to draw on other relevant literature. I felt the discussion was a little descriptive to be honest and this was where some of the progression data was introduced which came as a surprise after the findings focussed on the qualitative data? I wonder if you could take a look at this section and reference a little more literature and bring in some debate around NHS culture generally and perhaps supporting those earlier in their careers as identified earlier in the paper to draw this all together?

Response:

Thank you for this suggestion. The discussion section has been reworked to include closer engagement with relevant literature and is now much stronger as a result.

R2 Comment:

I also felt the conclusion didn't accurately summarise/reflect what you described in your paper and missed the opportunity to make some bold suggestions for what we need to do next!

Response:

The conclusion has been reworded to better reflect the contents of the paper and to incorporate suggestions for improvement.

R2 Comment:

I hope the comments above are helpful to you, I really enjoyed reading your paper, we need more literature like this out there to support our work in this pathway....I look forward to seeing it published.

Response:

We appreciate this detailed feedback and feel that the manuscript is much improved thanks to the reviewer's suggestions.

Reviewer 3 Comment:

Good work adding to the body of literature providing the evidence that the clinical academic pathway for NMAHPs needs urgent attention to be equivalent to that of the medical CA pathway.

Response:

Thank you for this very positive feedback. A comparison study with medical clinical academics is currently underway.

VERSION 2 – REVIEW

REVIEWER	Professor Sheree Smith Lung, Sleep and Heart Health Research Network, School of Nursing and Midwifery, Western Sydney University Sydney AUSTRALIA
REVIEW RETURNED	16-Aug-2019

GENERAL COMMENTS	Thank you for taking the time to read our previous review comments and responding positively. The manuscript conveys an important message and offers some salient points for organisations that may wish to provide the best possible evidence based care and positive patient outcomes due to highly skilled nurses, midwives and allied health staff. This is a worldwide health service issue however, I understand the authors focused on their own geographical area as it is within their remit to have an impact. Until we have clinical professors with a label of their specialism (as does every other profession) to act as very visible role models, the process of reforming the NHS around clinical research careers will continue to be very slow. I hope your research career support network continues to grow and is replicated across England.
---

REVIEWER	Linda Tinkler Newcastle upon Tyne Hospitals NHS Foundation Trust UK
REVIEW RETURNED	11-Sep-2019

GENERAL COMMENTS	Thank you for responding positively to my initial comments. I have very much enjoyed reviewing this paper again and I think the amendments have made what was already a great piece of work much stronger. It was a pleasure to read and I am delighted to see this work being published... we need much more of this work to address concerns around this agenda!
--